# Inhibition of Protein Synthesis Attenuates Formation of Traumatic Memory and Normalizes Fear-Induced c-Fos Expression in a Mouse Model of Posttraumatic Stress Disorder

**DOI:** 10.3390/ijms25126544

**Published:** 2024-06-13

**Authors:** Tatyana A. Zamorina, Olga I. Ivashkina, Ksenia A. Toropova, Konstantin V. Anokhin

**Affiliations:** 1Institute for Advanced Brain Studies, Lomonosov Moscow State University, 119991 Moscow, Russia; tatazamorina@gmail.com (T.A.Z.); oivashkina@gmail.com (O.I.I.); xen.alexander@gmail.com (K.A.T.); 2Faculty of Biology, Department of Higher Nervous Activity, Lomonosov Moscow State University, 119234 Moscow, Russia; 3Laboratory of Neuronal Intelligence, Lomonosov Moscow State University, 119991 Moscow, Russia

**Keywords:** posttraumatic stress disorder, fear conditioning, c-Fos, protein synthesis, cycloheximide

## Abstract

Posttraumatic stress disorder (PTSD) is a debilitating psychosomatic condition characterized by impairment of brain fear circuits and persistence of exceptionally strong associative memories resistant to extinction. In this study, we investigated the neural and behavioral consequences of inhibiting protein synthesis, a process known to suppress the formation of conventional aversive memories, in an established PTSD animal model based on contextual fear conditioning in mice. Control animals were subjected to the conventional fear conditioning task. Utilizing c-Fos neural activity mapping, we found that the retrieval of PTSD and normal aversive memories produced activation of an overlapping set of brain structures. However, several specific areas, such as the infralimbic cortex and the paraventricular thalamic nucleus, showed an increase in the PTSD group compared to the normal aversive memory group. Administration of protein synthesis inhibitor before PTSD induction disrupted the formation of traumatic memories, resulting in behavior that matched the behavior of mice with usual aversive memory. Concomitant with this behavioral shift was a normalization of brain c-Fos activation pattern matching the one observed in usual fear memory. Our findings demonstrate that inhibiting protein synthesis during traumatic experiences significantly impairs the development of PTSD in a mouse model. These data provide insights into the neural underpinnings of protein synthesis-dependent traumatic memory formation and open prospects for the development of new therapeutic strategies for PTSD prevention.

## 1. Introduction

Posttraumatic stress disorder (PTSD) is an incapacitating condition characterized by compromised emotional reactions resulting from acute psychological trauma. The disorder is marked by the impairment of information processing in fear memory brain circuits and the establishment of strong memories associated with the traumatic episode [1,2,3]. Memory of a traumatic event is implicated in many symptoms of PTSD, including intrusive thoughts and dreams, obsessive memories, flashbacks, and abnormal avoidance behavior [4,5,6]. Obsessive memories in PTSD patients are unusually strong, with reliving experiences comparable in intensity to the original trauma (flashbacks) [4]. However, PTSD is distinguished not only by extraordinarily vivid memories of certain aspects of a traumatic event (hypermnesia) but also by partial retrograde amnesia for other aspects [7]. This evidence has led to the hypothesis of aberrant aversive memory formation following a traumatic event. Support for this hypothesis is found in the impaired extinction of traumatic memories in PTSD [8,9,10], unlike normal aversive memories formed through fear conditioning, which can be extinguished [11]. fMRI studies have also demonstrated altered activation of fear memory neural circuits in PTSD patients [12,13,14,15,16,17,18], including the amygdala [12,13], hippocampus [14,15], cingulate [16,17], and prefrontal cortices [17,18]. 

To explain PTSD aberrant memories, Siegmund and Wotjak [19] proposed a “dual-branch hypothesis,” which suggests that traumatic experiences simultaneously lead to associative memories (involving classical conditioning) and non-associative memories (involving sensitization processes) [19,20]. Building on this hypothesis, authors developed a mouse PTSD model that replicates key features of the traumatic event leading to PTSD in humans: unpredictability and uncontrollability [20]. The model exhibits a good alignment of animal behavioral phenotypes with human PTSD symptoms, including prolonged PTSD manifestations and impaired traumatic memory extinction [19,20]. The developed mouse model, based on the presentation of a robust and unavoidable footshock, allows experimental study of both associative and sensitized PTSD components. While the associative component undergoes typical consolidation stages mirroring normal aversive memory, the sensitization processes unfold more slowly, requiring an incubation period for its expression [20]. Distinct synaptic mechanisms underlie the two components: associative memory is N-Methyl-D-Aspartate (NMDA)-dependent, whereas sensitized components are not [21]. However, the brain activity in this PTSD model has not been studied, leaving its validation for human PTSD symptoms and mechanisms incomplete. Therefore, in the present study, we applied brain-wide c-Fos imaging [22,23] to compare brain activity patterns after retrieval of traumatic and normal fear memory in this experimental model. 

We also explored the effects of inhibition of protein synthesis (PSI) on the formation of associative and non-associative components of traumatic memory and concomitant patterns of brain activity. Protein synthesis inhibitors (PSIs) administered before or shortly after the traumatic event are known to prevent the development of PTSD-like behavioral symptoms [24,25,26]. These data suggest a dependency of PTSD on molecular mechanisms involved in the consolidation of normal aversive memory [27,28]. Still, it remains unclear whether such relief of behavioral PTSD-like symptoms is accompanied by normalization of brain activity in fear circuits. By administering the PSI cycloheximide before traumatic events, we investigated its effects on associative (conditioned fear) and non-associative (behavioral sensitization) PTSD-like symptoms and patterns of brain activity related to the retrieval of intact or impaired fear and traumatic memory. 

## 2. Results

### 2.1. Effects of Protein Synthesis Inhibition on PTSD-Induced Fear and Anxiety

In this experiment, we studied the effect of PSI cycloheximide (CXM, 100 mg/kg) on PTSD formation in mice in a robust footshock model [20,29]. We used five groups of animals: mice trained in a standard context fear conditioning (FC) paradigm with CXM (FC + CXM, n = 9) or saline (FC + Sal, n = 11) injection; PTSD mice with CXM (PTSD + CXM, n = 15) or saline injection (PTSD + Sal, n = 12) and an active control group with saline injection (No shock, n = 15). For FC training, mice were placed in a novel context, allowed to explore for a while, and then given a 1 mA footshock for 2 s. It is known that such a moderate footshock induces normal aversive fear memory in mice [29]. For PTSD induction, mice were placed in the same context, allowed to explore, and then given three strong footshocks of 1.5 mA, each lasting 10 s. We previously demonstrated that such a procedure induces a PTSD-like phenotype in mice [29]. Control No shock mice were only exposed to the context without receiving footshock. All animals were then returned to the shocked context forassociative fear memory test. All animals underwent the following experimental stages: (1) FC training/PTSD induction/context exposure; (2) conditioned (associative) fear test; tests for non-associative consequences of the traumatic event—(3) behavioral sensitization and (4) anxiety in the elevated plus-maze (Figure 1a).

Freezing behavior was nearly absent in all animals during the novel context exploration before the footshock (Figure 1b). All animals that received the footshock demonstrated significantly more freezing behavior immediately after the footshock compared to the episode before footshock application (two-way ANOVA, post hoc Sidak’s multiple comparisons test, factor “Group”: F (4, 57) = 95.75, *p* < 0.0001; factor “Time interval”: F (1, 51) = 50.85, *p* < 0.0001; interaction of factors: F(4, 57) = 97.55, *p* < 0.0001; pairwise comparison of intervals before and after footshock: *p* < 0.0001 for FC + Sal, FC + CXM, PTSD + Sal, PTSD + CXM groups). All animals that received footshock also demonstrated significantly more freezing behavior immediately after footshock than the control group animals (Tukey’s multiple comparisons test, *p* < 0.0001). Finally, the freezing level after the footshock application in the PTSD + CXM and PTSD + Sal groups was significantly higher compared to the FC + Sal and FC + CXM groups (*p* < 0.0001). Thus, administering saline or CXM did not affect the level of fear in naive mice in a novel context. Footshock in this context caused the formation of short-term fear memory for both moderate and robust aversive stimuli. At the same time, the strong footshock used for PTSD induction produced a higher level of freezing than the moderate footshock used in FC training.

In the associative memory retrieval test seven days after the training, the animals of the FC + Sal and PTSD + Sal groups showed a significantly higher level of freezing than the control animals (one-way ANOVA: F (4, 57) = 87.81, *p* < 0.0001, post hoc Tukey’s multiple comparisons test: *p* < 0.0001). The freezing level in the PTSD + Sal group was significantly higher than in FC + Sal mice (*p* = 0.0032) (Figure 1c). The CXM injection before the FC training or PTSD induction resulted in impairment of associative memory, both normal and traumatic. Thus, the animals of the FC + CXM group showed significantly less freezing behavior than the mice that received a saline injection in the FC + Sal group (*p* < 0.0001) and did not differ in the level of freezing from the control animals (*p* = 0.8880). At the same time, animals of the PTSD + CXM group showed significantly less freezing behavior than PTSD + Sal group mice (*p* < 0.0001) and even less than animals with normal memory (PTSD + Sal vs. FC + Sal: *p* = 0.0018), but significantly more than the control animals (*p* < 0.0001).

PTSD animals exhibited higher levels of freezing during the memory retrieval compared to the animals with normal aversive memory. Injection of CXM before memory formation led to the complete erasure of the contextual fear in the FC paradigm. However, CXM injection before the PTSD induction did not cause a complete disruption of the fear memory, though it significantly reduced freezing levels compared to the saline group. Notably, the level of fear in animals exposed to CXM before the PTSD induction was relatively high, although significantly lower than in groups of animals with normal aversive memory. Thus, the protein synthesis inhibition before the traumatic exposure prevented the formation of a robust associative fear in animals with PTSD but did not lead to the complete elimination of the contextual fear memory, as in animals that were exposed to usual contextual fear conditioning.

Next, we performed a behavioral sensitization test (Figure 1d). In a novel context (before sound presentation), the level of freezing in the groups with normal aversive memory did not differ from that in the control animals (two-way ANOVA, post hoc Tukey’s multiple comparisons test, factor “Group”: F (4, 57) = 25.62, *p* < 0.0001; factor “Time interval”: F (1, 57) = 9.260, *p* = 0.0035; interaction of factors: F (4, 57) = 5.224, *p* = 0.0311; FC + Sal or FC + CXM vs. No shock: *p* = 0.1344). Thus, there was no generalization of aversive memory to the new context in trained mice. The PTSD + Sal group had a significantly higher level of freezing in the new context compared to the control group and the FC + Sal group (when comparing PTSD + Sal and No shock groups: *p* < 0.0001, when comparing PTSD + Sal and FC + Sal groups: *p* = 0.0002). At the same time, CXM administration led to a reduced level of freezing in the new context in the PTSD animals (*p* < 0.0001, compared to the PTSD + Sal group), and the level of freezing in the new environment did not differ in the PTSD + CXM and No shock groups (*p* = 0.3582). 

The sensitized fear component test showed that fear conditioning did not affect freezing to a novel sound compared to a control group (*p* = 0.1593), i.e., FC + Sal animals did not demonstrate behavioral sensitization. The freezing level in the PTSD + Sal group was higher than in the other groups during the presentation of a novel sound (PTSD + Sal vs. No shock: *p* < 0.0001; PTSD + Sal vs. FC + Sal: *p* < 0.0001; PTSD + Sal vs. PTSD + CXM: *p* < 0.0001) (Figure 1d), indicating that sensitization developed in mice with PTSD. When comparing the freezing level before and after the sound presentation, a significant increase in freezing behavior was observed only in the PTSD + Sal group (Sidak’s multiple comparisons test, freezing level in PTSD + Sal mice before and during a sound presentation: *p* = 0.03). CXM administration led to a significant decrease in the freezing level in the PTSD animals (*p* < 0.0001), and the freezing level of the PTSD + CXM group during novel sound presentation did not differ from that of the control animals (*p* = 0.9840). These results indicate that protein synthesis inhibition during the induction of PTSD prevented the development of a sensitized fear.

To evaluate the level of anxiety, we analyzed animal behavior in the elevated plus-maze. Animals in the FC + Sal and FC + CXM groups did not differ from the control group in time spent in the open arms (one-way ANOVA, post hoc Tukey’s multiple comparisons test: F (4, 45) = 11.38, *p* < 0.0001; FC + Sal or FC + CXM vs. No shock: *p* = 0.9999), i.e., they did not exhibit elevated levels of anxiety (Figure 1e). At the same time, mice of the PTSD + Sal group spent less time in open arms compared with animals of the No shock and FC + Sal groups (PTSD + Sal vs. No shock: *p* = 0.0202; PTSD + Sal vs. FC + Sal: *p* = 0.032). Animals in the PTSD + CXM group spent more time in open arms compared to the animals in the No shock (*p* = 0.0081), FC + Sal (*p* = 0.0172), and PTSD + Sal groups (*p* < 0.0001).

The total freezing time in the FC + Sal (*p* < 0.0001), FC + CXM (*p* = 0.0042), and PTSD + Sal (*p* < 0.0001) groups was significantly higher compared to the control animals (F (4, 45) = 14.79, *p* < 0.0001, Figure 1f). At the same time, animals in the PTSD + CXM group demonstrated less freezing behavior compared to the PTSD + Sal (*p* = 0.0003) and FC + Sal (*p* = 0.0009) groups and did not differ in their level of freezing from the control group (*p* = 0.4872).

All groups of animals that received footshock had a lower number of entries into the closed arms compared with the control group (F (4, 45) = 4. 426, *p* = 0.0042; when compared with the FC + Sal group: *p* = 0.0069, FC + CXM: *p* = 0.0214, PTSD + Sal: *p* = 0.0241, PTSD + CXM: *p* = 0.0401, Figure 1g). We found no significant differences between the groups for the percentage of time in the center (F (4, 45) = 0.9539, *p* = 0.4420). So, protein synthesis inhibition during PTSD induction prevented increased anxiety in animals.

### 2.2. Patterns of c-Fos Brain Activity following Fear and Traumatic Memory Retrieval and the Effects of Protein Synthesis Inhibition

Next, we compared patterns of brain activity in animals during the retrieval of normal fear or traumatic memories, both disrupted and undamaged by CXM injection.

The formation of traumatic memory was preceded by the administration of CXM, while the control group received an injection of saline. There were four groups in the experiment: FC + Sal (n = 13), PTSD + Sal (n = 13), PTSD + CXM (n = 14), and Home cage (animals sacrificed from the home cage, n = 8). Seven days after PTSD induction or FC training, we retrieved normal or traumatic memory, and then brain samples of the animals were taken—Figure 2a. 

We found a higher activation level in the lateral (*p* = 0.0201, Figure 2b) [LA] and basolateral (*p* = 0.0492, Figure 2c) [BLA] amygdala nuclei, prelimbic (*p* = 0.0225, Figure 2j) [PrL] and cingulate (*p* = 0.006, Figure 2l) [Cg] cortices and paraventricular nucleus of the thalamus (*p* = 0.0028, Figure 2i) [PV] in FC + Sal group compared to Home cage group—one-way ANOVA, post hoc Tukey’s multiple comparisons test: LA: F (3, 42) = 3.261, *p* = 0.0307; BLA: F (3, 41) = 8.584, *p* = 0.0002; PrL: F (3, 42) = 4.647, *p* = 0.0068; Cg: F (3, 43) = 9.255, *p* < 0.0001; PV: F (3, 40) = 16.03, *p* < 0.0001.

We found a significant increase in the number of c-Fos-positive cells in the PTSD + Sal group after the traumatic memory retrieval compared to the Home cage group in the infralimbic (*p* < 0.0001, Figure 2k) [IL], PrL (*p* = 0.006, Figure 2j) and Cg cortices (*p* = 0.0004, Figure 2l); in the CA3 field of the hippocampus (*p* = 0.0307, Figure 2g); BLA (*p* = 0.0028, Figure 2c); and PV (*p* < 0.0001, Figure 2i)—one-way ANOVA, post hoc Tukey’s multiple comparisons test: BLA: F (3, 41) = 8.584, *p* = 0.0002; PL: F (3, 42) = 4.647, *p* = 0.0068; IL: F (3, 37) = 12.46, *p* < 0.0001; Cg: F (3, 43) = 9.255, *p* < 0.0001; CA3: F (3, 38) = 5.802, *p* = 0.0023; PV: F (3, 40) = 16.03, *p* < 0.0001. We also showed a significantly higher level of activation of the IL (*p* = 0.0036) and PV (*p* = 0.015) in the PTSD + Sal group compared to the FC + Sal group, indicating enhanced activation of these areas, specifically during traumatic but not normal aversive memory retrieval.

Injection of CXM before the traumatic experience resulted in the decreased number of c-Fos-positive cells after the retrieval of memory in PTSD + CXM animals compared to the intact traumatic memory retrieval in PTSD + Sal group in the Cg (*p* = 0.0188, Figure 2l); the CA3 (*p* = 0.0114, Figure 2g); BLA (*p* = 0.0006, Figure 2c); central amygdala nucleus (*p* = 0.0034, Figure 2d) [CeA]; PV (*p* = 0.0022, Figure 2i) and periaqueductal gray matter (*p* = 0.0395, Figure 2e) [PAG]—one-way ANOVA, post hoc Tukey’s multiple comparisons test: Cg: F (3, 43) = 9.255, *p* < 0.0001; CA3: F (3, 38) = 5.802, *p* = 0.0023; BLA: F (3, 41) = 8.584, *p* = 0.0002; CeA: F (3, 41) = 4.936, *p* = 0.0051; PV: F (3, 40) = 16.03, *p* < 0.0001; PAG: F (3, 40) = 5.345, *p* = 0.0034. Furthermore, animals that received an injection of CXM before PTSD induction did not show differences from the Home cage group in the vast majority of structures examined, except for the IL (*p* = 0.0071, Figure 2k) and PV (*p* = 0.0038, Figure 2I)—one-way ANOVA, post hoc Tukey’s multiple comparisons test: IL: F (3, 37) = 12.46, *p* < 0.0001; PV: F (3, 40) = 16.03, *p* < 0.0001. A lower number of c-Fos-positive cells compared to the FC + Sal group was observed in animals of the PTSD + CXM group in the Cg (*p* = 0.03, Figure 2i); CA3 (*p* = 0.0337, Figure 2g); BLA (*p* = 0.0239, Figure 2c); and PAG (*p* = 0.0059, Figure 2k)—one-way ANOVA, post hoc Tukey’s multiple comparisons test: Cg: F (3, 43) = 9.255, *p* < 0.0001; CA3: F (3, 38) = 5.802, *p* = 0.0023; BLA: F (3, 41) = 8.584, *p* = 0.0002; PAG: F (3, 40) = 5.345, *p* = 0.0034. Thus, preventing the development of traumatic memory through PSI injection resulted in the normalization of behavior and restoration of normal patterns of brain c-Fos activity.

## 3. Discussion

In the first experiment, we investigated the effect of protein synthesis inhibition during traumatic experience on subsequent PTSD-like behavioral manifestations. Although protein synthesis inhibitors can produce acute effects on neural activity and behavior [30,31,32], administering CXM 30 min before the footshock did not affect the subsequent freezing behavior of mice in the conditioning chamber. This result suggests that the observed action of CXM is not due to its nonspecific effects on mice reactivity during PTSD acquisition.

Moderate and strong footshock both led to the formation of contextual fear memory. Higher-intensity footshock during PTSD induction resulted in increased fear levels compared to animals with normal aversive memory. Protein synthesis inhibition by CXM entirely prevented normal fear memory. The animals showed no memory of the aversive experience seven days after training, with fear levels similar to the control group without footshock. The effect of CXM on traumatic memory was less pronounced. In mice receiving CXM before the traumatic event, the levels of contextual fear were reduced compared to animals with traumatic and normal aversive memory. However, the contextual fear memory was not entirely eliminated since freezing did not decrease to the level of the control group. Thus, exposure to a traumatic event in an animal model of PTSD leads to the formation of a persistent memory, with the associative fear memory component proving resistant to such strong disruptive influences as protein synthesis inhibition, a phenomenon not previously reported [24,25,26]. 

When exposed to novel contexts or sound stimuli, mice with PTSD exhibited elevated fear levels, a characteristic of the sensitization of the fear response seen in PTSD. Remarkably, protein synthesis inhibition before the footshock completely disrupted these PTSD-like behavioral manifestations when assessed seven days after training.

Further analysis of anxiety-related behaviors and overall activity in the elevated plus maze revealed significant differences between PTSD animals and those with impaired traumatic memory. PTSD mice displayed increased freezing behavior, reduced time spent in open arms, and diminished exploratory behavior. These changes indicated substantial alterations in fear and defensive behavior in potentially threatening environments, consistent with the development of PTSD symptoms. However, protein synthesis inhibition before traumatic memory formation led to the normalization of elevated plus-maze behavior. Intriguingly, animals with impaired traumatic memory spent even more time in the maze’s open arms than those without any footshock experience, suggesting that traumatic memory impairment by protein synthesis inhibition may induce complex changes in stress response systems due to prolonged CXM effects.

Contextual fear conditioning also influenced the anxiety levels of animals. Those with normal aversive memory exhibited a slight increase in anxiety, reflected in increased freezing duration and reduced exploratory activity in the elevated plus maze, indicating an increase in anxious behavior. Importantly, protein synthesis inhibition before the formation of normal aversive memory had no impact on anxiety or exploratory activity, with animals having impaired and preserved normal fear memory displaying similar elevated plus maze performance.

In summary, these results demonstrate that strong, inescapable footshock leads to the formation of robust contextual associative memory and significant reorganization of fear and defensive behavior in mice, resulting in the manifestation of PTSD-like behaviors. Protein synthesis inhibition before regular contextual fear conditioning results in the complete prevention of long-term memory consolidation, while administration before the strong footshock impairs but does not entirely prevent the development of PTSD symptoms. Administration of CXM during PTSD induction almost completely eliminates symptoms of sensitization and anxiety without erasing the association between the context and the footshock.

We therefore hypothesize that there is an early protein-dependent stage in the development of PTSD, akin to the consolidation stage of long-term memory formation [24,25,33,34]. This stage involves the formation of long-term associative traumatic memory and protein-dependent non-associative plasticity within neuronal fear and defense circuits. It may explain the need for an incubation period to develop hyperreactivity symptoms [21], as initial non-associative plasticity processes may be necessary but insufficient for full PTSD manifestation. Subsequent, slow development and modification of non-associative plasticity processes may involve new neuronal populations analogous to processes of distant memory consolidation [35,36,37,38]. This process could be triggered by intrusive memories in humans or circumstantial reminders in rodents [39]. 

Previous studies have explored the impact of protein synthesis inhibition on PTSD development, primarily using the rat model of acute predator stress and without investigating associative memory formation [24,25,33,40]. Our findings align with prior research, indicating that protein synthesis inhibition during traumatic experiences can suppress PTSD symptoms. However, our data suggest that reorganization in fear and defense neuronal systems during PTSD development is not unidirectional. Following a traumatic experience, profound protein-dependent reorganization occurs at the neuronal level, disrupting typical defensive behavioral patterns. These changes result in a reduced anxiety level compared to animals without any traumatic experience. Our data suggest the possibility that old, unreactivated memories may be impaired during the consolidation of new memories [41]. Moreover, it raises the question of whether the innate behavior could be impaired at its realization.

Using a PTSD induction model based on the footshock, a series of animal experiments were conducted to investigate the involvement of molecular mechanisms of aversive memory in the development of the disorder [42,43]. It was shown that at the moment of trauma, molecular cascades involved in the development of NMDA-dependent long-term potentiation are activated in the hippocampus and amygdala [44,45,46]. An increase in the phosphorylation level of CREB (calcium-response element binding protein) during trauma was also demonstrated [47,48,49], along with the activation of gene expression related to memory consolidation [26,50,51]. Meanwhile, administering kinase inhibitors involved in memory consolidation at the moment of trauma reduces the likelihood of developing PTSD in animals [2,52]. Furthermore, in another PTSD model involving predator exposure, it was shown that inhibition of protein synthesis disrupts the development of PTSD-like behavioral disorders [24,25,26] within the same period as memory formation [34] and leads to a decrease in corticosterone levels in the animals’ blood [33]. These findings support the hypothesis that the development of PTSD and memory consolidation in classical conditioning depend on similar molecular mechanisms.

Lastly, our examination of c-Fos expression patterns revealed significant activation of specific brain regions during the development of PTSD-like behavior, including the prefrontal cortex (PrL, IL, Cg), amygdala (BLA), and PV. This activation pattern is consistent with data on the involvement of these regions in PTSD in humans and animals [53,54,55,56]. The high BLA activity in PTSD aligns with extensive evidence linking amygdala functioning to PTSD symptoms in both humans [57,58,59] and animals [60,61]. For instance, an NMDA receptor antagonist in the amygdala reduced anxiety symptoms in a predator-exposure-based PTSD model [61]. Cg activity is associated with fear expression in the contextual FC paradigm [62]. Abnormal Cg activity may reflect impaired emotional regulation, increased reactivity to aversive stimuli, and fear generalization in PTSD [63,64,65]. IL dysfunction is linked to PTSD, as deficits in memory extinction accompany the disorder [66,67]. IL is central to fear memory extinction [68,69,70,71]. Increased c-Fos expression in IL during traumatic memory retrieval may indicate abnormal plasticity associated with impaired extinction. Elevated PV activity in mice with PTSD aligns with its role in regulating fear behavior [72,73] and heightened sensitized fear components.

Protein synthesis inhibition before traumatic experience significantly impacted brain activation patterns after memory retrieval. Reduced c-Fos-positive cell numbers in BLA, CeA, PAG, CA3, PV, IL, and Cg in the PTSD + CXM group indicated decreased activity in these structures. Lower BLA activity may reflect reduced hyperreactivity to dangerous stimuli and reduced anxiety [74]. In addition, the reduced activity of this structure may be reflected in reduced anxiety in mice compared to the group with intact traumatic memory [12,75]. Reduced CeA activation suggests a less pronounced fear response to trauma context, as CeA mediates emotional responses and fear extinction [57,76,77]. PAG, linked to fight or flight responses, showed abnormal activity in PTSD, leading to increased fear expression [5,78]. Thus, decreased PAG activity in the PTSD + CXM group may signify normalized fear and anxiety reactions. Lower CA3 activity suggests more accurate context memory encoding, which was observed as improved discrimination between dangerous and safe contexts in animals with impaired traumatic memory. Normalized Cg activity may indicate reduced generalization in mice with impaired PTSD [76], contributing to behavior more similar to control animals without traumatic experience. 

At the molecular level PTSD is characterized by a complex interplay of various signaling pathways that contribute to its induction and development. Molecular mechanisms involved in PTSD are diverse and encompass changes in neurotransmitters, receptors, neurotrophic factors, and intracellular signaling cascades [52,79]. Signaling pathways involved in PTSD include those mediated by cyclic AMP response element-binding protein (CREB) and mitogen-activated protein kinases (MAPKs) [79]. One or several of these described molecular mechanisms may be responsible for the normalization of brain activity patterns during memory retrieval.

Altogether, the c-Fos activity patterns obtained in our study are consistent with the existing literature linking the discussed brain regions to PTSD in both humans and animals. However, c-Fos imaging is a correlative approach. Further exploration of the neural substrates involved in PTSD and protein synthesis-dependent consolidation using approaches for the causal investigation such as optogenetic manipulation will be critical to understanding this complex disorder and potential therapeutic strategies.

## 4. Materials and Methods

### 4.1. Animals

Male C57Bl/6J mice (3–4 months old) were used. Mice were group-housed 3–5 per cage with the ad libitum food and water access. Animals were kept under a 12 h light/dark cycle. Experiments were performed during the light phase of the cycle. All methods for animal care and all experiments were approved by the Commission on Bioethics of Lomonosov Moscow State University (Application No. 178-a, approved during the Bioethics Commission meeting No. 162-d held on 16 May 2024) and were in accordance with the Russian Federation Order Requirements N 267 M3 and the National Institutes of Health Guide for the Care and Use of Laboratory Animals.

### 4.2. Behavior

During FC training, mice were placed into the fear conditioning chamber (MED Associates Inc., Fairfax, VT, USA) for a 230 s exploration of context A. Then, the footshock (2 s, 1 mA—moderate footshock) was applied, and animals remained in context A for 128 s before being returned to their home cages. PTSD induction was performed according to the modified protocol by Siegmund and Wotjak [20,29]. For PTSD induction, mice were placed in context A for 170 s, then three consecutive footshocks were applied (10 s, 1.5 mA—strong footshock) v2.5.5.0, and animals remained in the chamber for 60 s. The interval between the footshocks was 50 s. Animals of the No shock group were placed in context A for 300 s without footshock.

Seven days later, animals were placed in context A for context memory retrieval (180 s). One week is a necessary interval for the development of posttraumatic behavioral symptoms [20]. The following day (the eighth day after the FC training/PTSD induction), animals were placed in context B for a memory sensitization test [20]. After 180 s of novel context exploration, a neutral tone (80 dB, 9 kHz) was presented for 180 s.

Context A was a white light (87 lux) and IR light illuminated plastic box (30 cm × 23 cm × 21 cm) with a grid floor. To change the context for the memory sensitization test, we placed the black plastic A-shaped insert into the chamber and covered the grid floor with a plastic sheet and wood sawdust (context B). Context B was illuminated with an IR light. Context A and B were cleaned before and after each session with 70% ethanol or 53% ethanol solution of peppermint oil, respectively. Freezing behavior was scored automatically using Video Freeze software version No. 2.5.5.0 (MED Associates Inc., Fairfax, VT, USA). 

On the ninth day after the FC training/PTSD induction, animals were tested in the elevated plus-maze to address the anxiety level [80,81,82]. Animal behavior in the elevated plus-maze was video recorded and analyzed using the EthoVision XT software version No. 8.5 (Noldus, Wageningen, The Netherlands). The following behavioral parameters were analyzed: (i) time spent in the open arms and (ii) freezing duration (as measures of anxiety [83,84,85]), (iii) time spent on the central platform (as a measure of risk assessment and decision making [86,87,88]), (iv) the number of entries to the closed arms (as a measure of exploratory behavior [84]).

### 4.3. Cycloheximide Injection

Animals received a single i.p. injection of CXM (100 mg/kg, Sigma-Aldrich, St. Louis, MO, USA) 30 min before the FC training/PTSD induction. At this dose, CXM suppresses protein synthesis in the mouse brain at 95% level 10–40 min after administration [89,90]. Control groups of animals received a single i.p. injection of saline 30 min before the FC training/PTSD induction.

### 4.4. Immunohistochemistry

To analyze the brain’s activity, mice were sacrificed by cervical translocation 90 min after context memory retrieval. Home cage control animals with no novel experience were sacrificed at the same time. Brains were removed and immediately frozen in liquid nitrogen vapors. The 20 µm coronal sections were cut in a cryostat (Leica CM1950, Leica Biosystems, Wetzlar, Germany) and fixed in 4% paraformaldehyde. Brain sections were taken at various distances from the Bregma [91]: (i) +1.7 mm for the infralimbic (IL) and the prelimbic (PrL) cortices; (ii) +0.98 mm for the cingulate (Cg) cortex; (iii) −1.46 mm for the retrosplenial (RS) cortex, hippocampal CA1, CA3, and dentate gyrus (DG), the basolateral (BLA), lateral (LA) and central (CeA) amygdalar nuclei and for the paraventricular (PV) thalamic nucleus; (iv) −3.15 for the periaqueductal gray matter (PAG). For c-Fos immunostaining, primary rabbit polyclonal antibodies against the c-Fos protein (226 003, Synaptic Systems, Goettingen, Germany, dilution 1:1000) and donkey secondary antibodies against rabbit conjugated with Alexa 488 (A-21206, Thermo Fisher, Waltham, MA, USA, dilution 1:500) were used. Whole section images were acquired with the fluorescence microscope scanner (Olympus VS110, Waltham, MA, USA) at the 10×.

The c-Fos-positive cell count was performed using the Image-Pro Plus 3.0 software (Media Cybernetics, Rockville, MD, USA). The density of cells was calculated as the ratio of the number of positive cells to the structure area (in mm^2^). Three slices from each brain were analyzed for each structure.

### 4.5. Statistical Analysis

Statistical analysis was performed using the Prism 7 software (GraphPad, Boston, MA, USA). We used one-way and two-way ANOVA followed by multiple comparisons with Tukey’s post hoc for unpaired samples and Sidak’s post hoc for paired samples, with a significance level set to 0.05. Graphs were plotted in Prism, and schemes of experiments were created with BioRender.com, accessed on 3 October 2023.

## Figures and Tables

**Figure 1 ijms-25-06544-f001:**
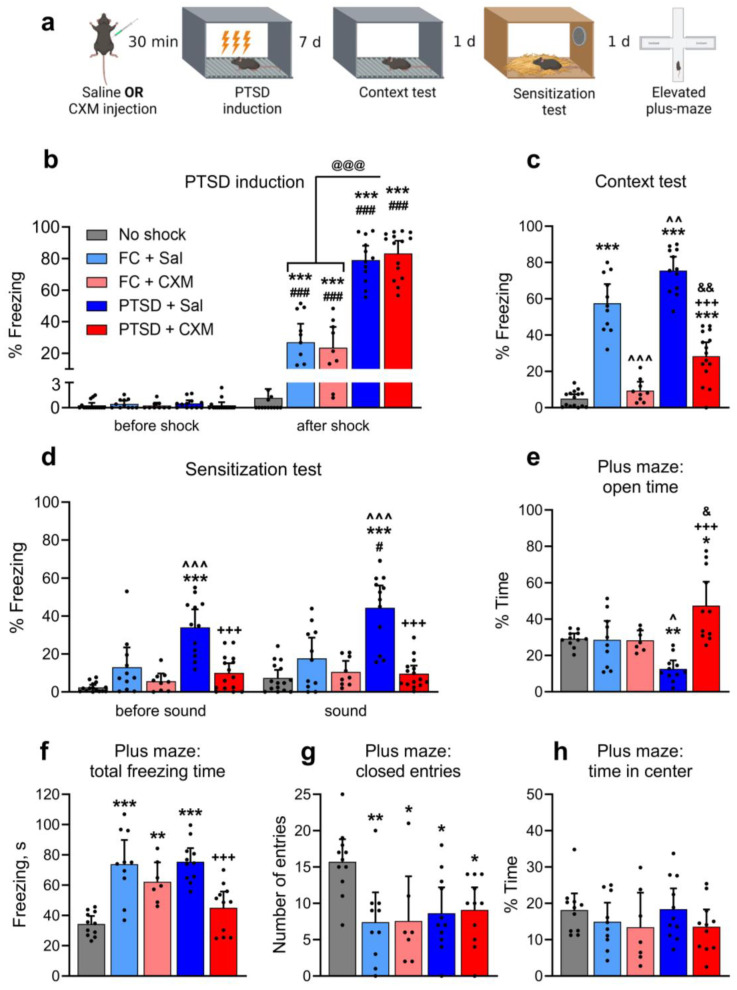
(**a**) Experimental design. Freezing level during (**b**) PTSD induction (3 × 1.5 mA, 10 s footshock in context A) or normal fear memory formation (1 × 1 mA, 2 s footshock in Context A), (**c**) context memory retrieval (3 min in context A) and (**d**) fear sensitization test (3 min in context B + 3 min sound 75 dB, 80 kHz), mean, 95% CI. Elevated plus-maze results: (**e**) time spent in open arms, (**f**) total freezing time in elevated plus-maze, (**g**) number of entries to the closed arms, (**h**) time spent in the center of elevated plus-maze, mean, 95% CI. *—*p* < 0.05, **—*p* < 0.01, ***—*p* < 0.0001, compared to the No shock group (n = 15); ^—*p* < 0.05, ^^—*p* < 0.01, ^^^—*p* < 0.0001, compared to the FC + Sal group (n = 11); &—*p* < 0.05, &&—*p* < 0.01, compared to the FC + CXM group (n = 9); +++—*p* < 0.0001, compared to the PTSD + Sal group (n = 12); #—*p* < 0.05, ###—*p* < 0.0001 compared to the freezing level in the same group in different time interval, @@@—*p* < 0.0001. FC + CXM group—n = 15. One-way or two-way ANOVA followed by multiple comparisons with Tukey’s post hoc for unpaired samples and Sidak’s post hoc for paired samples, with a significance level set to 0.05.

**Figure 2 ijms-25-06544-f002:**
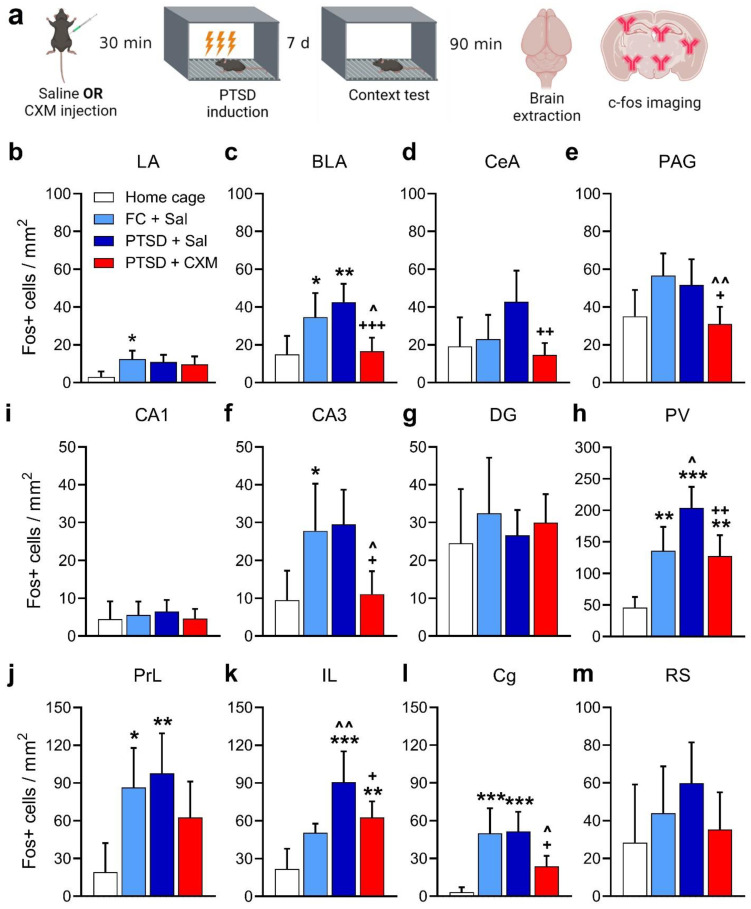
(**a**) Experimental design. (**b**–**m**) c-Fos brain activation after the intact or impaired memory retrieval (mean, 95% CI). BLA—basolateral nucleus of amygdala; CA1—hippocampal CA1; CA3—hippocampal CA3; CeA—central nucleus of amygdala; Cg—cingulate cortex; DG—dentate gyrus; IL—infralimbic cortex; LA—lateral nucleus of amygdala; PAG—periaqueductal gray matter; PrL—prelimbic cortex; PV—paraventricular thalamic nucleus; RS—retrosplenial cortex. *—*p* < 0.05, **—*p* < 0.01, ***—*p* < 0.001, compared to the Home cage group (n = 8); ^—*p* < 0.05, ^^—*p* < 0.01, compared to the FC + Sal group (n = 13); +—*p* < 0.05, ++—*p* < 0.01, +++—*p* < 0.001, compared to the PTSD + Sal group (n = 13). PTSD + CXM group—n = 14. One-way ANOVA followed by multiple comparisons with Tukey’s post hoc for unpaired samples, with a significance level set to 0.05.

## Data Availability

The raw data supporting the conclusions of this article will be made available by the authors on request.

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
