# Peer review of "Inhibition of Protein Synthesis Attenuates Formation of Traumatic Memory and Normalizes Fear-Induced c-Fos Expression in a Mouse Model of Posttraumatic Stress Disorder"

_ijms, 2024, doi:10.3390/ijms25126544_

Round 1

Reviewer 1 Report

Comments and Suggestions for Authors

In this  study, the effects of moderate and strong foot shock exposure to several behavioral measures and neural activity measured by cFos were investigated. Animals were further treated with an protein synthesis inhibitor, leading to a distinct reduction of PTSD-like symptoms". While the study is very interesting in general, in my opinion, some comments need to be addressed before publication.

Specific comments:

(1) Line 21-23: Do you mean "only" during PTSD induction?

(2) Since the methods part comes after the results part, I would like to recommend to shortly introduce the fear conditioning and the PTSD procedure (2-3 sentences either in the intro or at the beginning of the results).

(3) Statistics: Aren't the training procedures (FC/PTSD) and the treatments (veh/CXM) two different factors? The authors combine both factors to the factor "group".

(4) It would be nice to see the individual data points in the bar diagrams. This helps to evaluate the variability and distribution of the data. In this context it would be also interesting whether there were indivual (resilient) animals that did not develop "PTSD-like symptoms" despite of  the PTSD protocol.

(5) For me, it is unclear whether the differences found in cFOS expression are specific to the PTSD protocol or simply to the higher freezing levels of the animals. To address this question, one could check whether cFOS expression is correlated with freezing levels in the context test. If only higher freezing levels are the cause of the difference, one should observe similar slopes of the regression lines, while if there are some qualitative differences in the protocol, there should be difference slopes in the different groups.

(6) How is cFOS expression affected by CXM? Does decrease cFOS expression after CMS really means that the neurons were less active? Please discuss?

(7) Line 403: Please change to "mA" (as above in line 399)

(7) I don't understand line 348-350. Please consider rephrasing or explaining this idea in more detail.

Reviewer 2 Report

Comments and Suggestions for Authors

-       While the experimental design appears robust, there are concerns regarding the specificity of the protein synthesis inhibitor (PSI) used (cycloheximide) and its potential off-target effects. Including additional control groups with alternative PSIs or employing genetic knockout models could strengthen the specificity of the observed effects.

-       The timing of PSI administration (30 minutes before footshock) seems arbitrary and lacks rationale. Justification for this specific timepoint should be provided, considering the kinetics of protein synthesis inhibition and its potential impact on fear memory formation.

-       The footshock paradigm used for PTSD induction should be further described, including parameters such as intensity, duration, and frequency, to facilitate replication and comparison with other studies.

-       The interpretation of freezing behavior as a proxy for fear memory is common but may overlook nuances in anxiety-like behavior. Additional behavioral assays, such as fear extinction or conditioned place preference, could provide a more comprehensive assessment of fear memory dynamics.

-       The discussion of footshock intensity influencing PTSD-like symptoms is speculative and requires empirical support. Directly comparing the effects of moderate and strong footshocks on fear memory consolidation and extinction could clarify this relationship.

-       While c-Fos imaging provides valuable insights into neural activity patterns, the interpretation of these data should be cautious, considering the multifaceted nature of PTSD neurobiology. Correlation between c-Fos activation and specific behavioral phenotypes should be validated through complementary approaches, such as optogenetic manipulation or chemogenetic inhibition.

-       The discussion emphasizes the normalization of brain activity following PSI administration without considering potential compensatory mechanisms or alternative pathways contributing to PTSD pathology. Acknowledging the complexity of neural circuits underlying PTSD and exploring additional molecular markers could provide a more nuanced understanding of PSI effects.

-       The discussion lacks integration with existing literature on the molecular mechanisms of fear memory formation and PTSD pathophysiology. Comparative analysis with human studies and clinical trials investigating PSI or related interventions would strengthen the translational relevance of the findings.

-       The conclusion implies a causal relationship between PSI administration and PTSD prevention without addressing potential confounding factors or alternative interpretations of the results. Emphasizing the exploratory nature of the study and suggesting avenues for future research would enhance the paper's scientific rigor.

Comments on the Quality of English Language

Minor errors in punctuation, tense consistency, and subject-verb agreement occur sporadically

Reviewer 3 Report

Comments and Suggestions for Authors

In the manuscript entitled “Inhibition of Protein Synthesis Attenuates Formation of Traumatic Memory and Normalizes Fear-induced c-Fos Expression in a Mouse Model of Posttraumatic Stress Disorder” Zamorina et al. used posttraumatic stress disorder (PTSD) animal model based on contextual fear conditioning in mice to investigate the neural and behavioral consequences of inhibiting protein synthesis. The authors used protein synthesis inhibitor that impaired the development of PTSD in a mouse model. Overall, this study seems quite interesting, but the results are poorly presented and need to be improved before proceeding further. There are a few issues, which need to be addressed:

·         The results are not clearly presented, there are unnecessary conclusions and citations that should be moved to the discussion:

“Thus, PTSD animals exhibited higher levels of freezing during the retrieval test compared to the animals with normal aversive ….were exposed to usual contextual fear conditioning”.

“These data suggest that animals after PTSD induction had increased level…inhibition prevented the development of this symptom”.

“To evaluate the level of anxiety, we analyzed animal behavior…proportion of time on the central platform (animal decision-164 making and risk assessment)”.

“Altogether, fear conditioning training led to a minor increase in anxiety…of increased anxiety in animals after a traumatic situation”.

“Inhibiting protein synthesis in the brain has been shown…both disrupted and undamaged by CXM injection”.

“Our results show that the infralimbic cortex and thalamic paraventricular nucleus…the normalization of behavior and restoration of normal patterns of brain c-Fos activity”.

·         The first Figure is illegible, I suggest presenting the results from this Figure in two separate Figures

·         In Figure 2, the abbreviations should be explained alphabetically or in the order of the Figures

·         In the introduction, the abbreviation NMDA should be expanded

·         Line 95-96: is the number N correct in the "Time interval" factor?

Comments on the Quality of English Language

Minor editing of English language required.

Round 2

Reviewer 2 Report

Comments and Suggestions for Authors

The authors have meticulously addressed the comments provided by the reviewers and have made significant improvements to the scientific paper. By incorporating the reviewers' feedback, they have clarified key points, enhanced the robustness of their methodologies, and provided more comprehensive analyses. These revisions have not only strengthened the arguments and conclusions presented but also improved the overall readability and coherence of the manuscript. The thorough and thoughtful responses to the reviewers' suggestions have resulted in a more rigorous and well-rounded piece of research.

Reviewer 3 Report

Comments and Suggestions for Authors

The description of the results has still not been improved sufficiently.

Moreover, such a high plagiarism report indicates the lack of authenticity of the work.

Comments on the Quality of English Language

Minor editing of English language required.